# Epstein–Barr Virus Association with Breast Cancer: Evidence and Perspectives

**DOI:** 10.3390/biology11060799

**Published:** 2022-05-24

**Authors:** Claudia Arias-Calvachi, Rancés Blanco, Gloria M. Calaf, Francisco Aguayo

**Affiliations:** 1Programa de Virología, Laboratorio de Oncovirología, Instituto de Ciencias Biomédicas (ICBM), Facultad de Medicina, Universidad de Chile, Santiago 8380000, Chile; clauarias@ug.uchile.cl (C.A.-C.); rancesblanco1976@gmail.com (R.B.); 2Instituto de Alta Investigación, Universidad de Tarapacá, Arica 1000000, Chile; gmc24@cumc.columbia.edu; 3Center for Radiological Research, Columbia University Medical Center, New York, NY 10032, USA; 4Universidad de Tarapacá, Arica 1000000, Chile

**Keywords:** Epstein–Barr virus, breast cancer, carcinogenesis

## Abstract

**Simple Summary:**

Epstein–Barr virus (EBV) is a very ubiquitous and persistent virus present in ~90% of the world population. The infection is generally asymptomatic during the lifetime, though it can cause lymphoid tumors and carcinomas in some subjects. The role of EBV in breast cancer (BC) has yet to be determined. In this review, we present the historical background and scientific evidence regarding the presence and potential role of EBV in this malignancy and we propose possible molecular mechanisms. Knowledge of EBV´s role in BC will contribute to establishing prevention strategies, early detection, and control of this highly aggressive and prevalent malignancy.

**Abstract:**

Epstein–Barr virus (EBV) is an enveloped DNA virus that belongs to the gamma Herpesviridae family. The virus establishes a latent/lytic persistent infection, though it can be involved in cancer development in some subjects. Indeed, evidence supports an etiological role of EBV in undifferentiated nasopharyngeal carcinoma (NPC), a subset of gastric carcinomas and lymphomas. Additionally, EBV has been detected in breast carcinomas (BCs) although its role has not been established. In this review, we summarize epidemiological information regarding the presence of EBV in BC and we propose mechanistic models. However, additional epidemiological and experimental evidence is warranted to confirm these models.

## 1. Introduction

Breast cancer (BC) is the most prevalent malignancy in both developing and industrialized countries [1]. Long-term exposure to risk factors, variable socioeconomic status, lack of primary attention facilities, and genetic factors, all contribute to the burden of BC [2,3]. Women are the most affected gender, although male BC cases have also been reported in some regions [4,5,6]. In 2020, BC ranked first among the 36 most common cancer types in 159 countries with 2,261,419 cases, accounting for an overall incidence of 11.7% of diagnosed cases. High-income regions such as Australia/New Zealand showed the highest age-standardized incidence rate (95.5/100,000) while South Central Asia had the lowest (26.2/100,000) [1]. Despite the current efforts to prevent this disease, an annual increase of 2% is expected to occur by 2030 [7]. The 5-year relative survival rate varies depending on the geographic region [8]. In 2020, BC ranked fifth among the 36 most common cancer types in 110 countries with 684,996 deaths, representing 6.9% of total cancer deaths. Melanesia showed the highest age-standardized mortality rate (27.5/100,000) while Eastern Asia showed the lowest (9.8/100,000). These data reflect the worldwide impact that BC represents and highlight the necessity to overcome health care barriers and promote access to screening strategies for BC prevention, early detection, and treatment [1,9].

Substantial efforts have been made to understand BC’s etiology. Some risk factors have been identified, such as female gender, germline mutations in BRCA1 or BRCA2 genes, first degree relatives with positive BC history, obesity, oral contraceptive usage, early menarche, late menopause, elderly nulliparity, sedentary lifestyle, alcohol intake, tobacco smoking and ionizing radiation exposure [10]. In fact, a Swedish case–control study showed an increased risk of developing basal-like BC, but no other subtypes (OR 4.17, 95% CI: 1.89–9.21) in women with no breastfeeding background versus other nulliparity or breastfeeding individuals [11]. A 20-year follow-up study in 514 women showed that postmenopausal BC patients showing unfavorable lifestyle factors, including obesity, physical inactivity, alcohol use, cigarette smoking or hypertension, had larger tumors than those with no such unfavorable lifestyle factors (26.3 versus 12.3 mm, *p* = 0.023) [12]. A cohort study showed that BRCA-related mutations are associated with a more aggressive phenotype, leading to a worse BC-related outcome when compared to other BC susceptibility genes such as ATM, CHEK2, STK11, NBN, PTEN, TP53, and PALB2 [13]. Additionally, it has been suggested that viral infections may be associated with BC pathogenesis [14]. In 1936, Bittner discovered a virus, later called the mouse mammary tumor virus (MMTV), in milk coming from an inbred albino breeding mouse with mammary gland carcinoma whose offspring later developed the same tumors [15]. This finding suggested the possibility that a human retrovirus analogue of MMTV could also be related to BC. Furthermore, it was suggested that multiple viral infections may have a role in BC, since some viral sequences were found in human BCs [16]. Of interest, high-risk human papillomavirus (HR-HPV) infections have been proposed as candidates related to BC development [17]. Finally, Epstein–Barr virus (EBV), a very ubiquitous persistent virus, has been detected in BCs, and is associated with this cancer [18,19]. Here, we focus on the epidemiological and experimental evidence which proposes a role of EBV in a subset of BCs.

## 2. Breast Cancer Classification

BC is a very heterogeneous disease, showing a high phenotypic variability [20,21]. The breast anatomy comprises the glandular tissue where primary BC tumors commonly develop in the lobules (milk-producing glands) or the ducts (milk passages) and the stroma tissue formed by fatty and connective structures [22]. Histologically, BCs are classified into two groups according to the tumor growth pattern. Invasive ductal carcinomas (IDC) and invasive lobular carcinomas (ILC) are the most frequent and aggressive BC types (80% and 10%, respectively), whereas ductal carcinoma in situ (DCIS) and lobular carcinoma in situ (LCIS) are uncommon types, unable to spread to the adjacent stroma. Furthermore, there are less frequent BC types such as medullary, mucinous (colloid), tubular and inflammatory carcinomas [23]. BCs are hormone-dependent malignancies expressing three receptors in tumors: the estrogen receptor (ER), the progesterone receptor (PR), and the human epidermal growth factor receptor 2 (HER-2) [24,25]. Importantly, BC is molecularly classified into four subtypes according to hormone receptor expression and the Ki-67 index into luminal-like (A and B), normal-like, HER-2 enriched and basal-like (triple negative breast cancer, TNBC) carcinomas. [26]. Additionally, the Ki-67 index is a nuclear marker of cell proliferation [26,27]. Luminal A is the most common molecular BC subtype (60%), is characterized by high ER and/or PR levels, is HER-2-negative, and Ki-67 is expressed in <14% of cases. Importantly, these tumors show a low histological grade, and patients have good prognosis. The Luminal B subtype is present in 20% of BC, is characterized by high ER and/or PR levels, is HER-2-negative, and it expresses higher Ki-67 levels (>14%) than the Luminal A subtype. Indeed, these tumors are more likely to have a high histological grade with an aggressive phenotype and low survival rates. The HER-2 subtype represents approximately 20% of BCs. It is important to mention that ER-PR negative tumors express high levels of Ki-67 (>14%). When HER-2 is overexpressed, it contributes to the development and progression of some aggressive BCs. The TNBC subtype accounts for 8–37% of BCs. ER-PR-HER-2 receptors are negative with high levels of Ki-67 (>14%). It is characterized as a highly infiltrating tumor, leading to metastasis with a worse clinical outcome [28]. The hormone receptor status (positive or negative) in BC tumors is considered a prognostic molecular biomarker. Generally, BCs are more likely to be characterized as high grade and show worse prognosis when they are classified as hormonal negative receptors [29]. Additionally, the Ki-67 index is useful as a classifier for Luminal A and Luminal B tumors and as a recurrence-free survival tool in clinical practice [30].

## 3. Epstein–Barr Virus Structure and Replication Cycle

Epstein–Barr virus (EBV) is an enveloped virus with a linear double-stranded DNA genome of approximately 172 kb in length packed into an icosahedral nucleocapsid. Additionally, the virus harbors a structure called tegument, located between the envelope and the nucleocapsid [31,32,33]. Sequencing-based analysis has led to EBV classification into two types: EBV-1 and EBV-2 [34]. 

EBV is a well-adapted virus able to persist in human beings for a lifetime through latency establishment in memory B lymphocytes [35]. In fact, approximately 90% of the human population are EBV carriers [36]. EBV infection during early childhood generally is asymptomatic, although when infection occurs during adolescence or early adulthood, individuals can experience symptomatic infectious mononucleosis (IM) [37].

B lymphocytes and epithelial cells are highly susceptible and permissive to EBV infection [38,39]. Additionally, EBV replication involves both a lytic and latent form of infection. 

EBV lytic replication occurs in both B lymphocytes and epithelial cells in immunologically competent individuals, while EBV latent infection is assumed to be restricted to B-cells [38,39,40]. In the oropharynx, viral particles have access to Waldeyer´s ring to infect lymphocytes. The viral mayor membrane glycoproteins gp350/220 allow the attachment to the host receptor CR2/CD21 while gp42 allows EBV entry by an endocytosis-mediated mechanism [41,42,43]. The viral genome is transported to the host nucleus where it establishes as a multicopy intranuclear circular episome due to the expression of Epstein–Barr Nuclear Antigen 1 (EBNA-1) with no viral progeny produced [44]. B lymphocytes go through a developing process which originates in four latency programs characterized by the specific expression of EBV latent encoded genes. In naïve B cells, all EBV latent genes from latency III are expressed: EBERs (small non-coding RNA), BARTs (miRNA), EBNA-1, EBNA-2, EBNA-3A, EBNA-3B, EBNA-3C, EBNA-LP, LMP-1 (latent membrane protein), LMP-2A, and LMP-2B to transform naïve B cells in lymphoblasts. Next, these cells migrate to the germinal center to accomplish differentiation process where EBERs, BARTs, EBNA-1, LMP-1, LMP-2A, and LMP-2B are expressed (latency II). When memory B cells are proliferating, only EBERs, BARTs and EBNA-1 are expressed (latency I), while resting B memory cells repress viral gene expression to avoid immune recognition (latency 0) [45]. Finally, the last step of B-cell differentiation into plasma cells triggers viral reactivation (switch from latent to lytic infection) with a spread of virions through nasopharyngeal secretions [46] (Figure 1). 

In epithelial cells, it is suggested that EBV entry is achieved due to the attachment of the viral gB and the receptor-binding complex gH/gL to the host Ephrin A2 (EphA2) receptor, which allows EBV entry by membrane fusion [47,48]. The viral genome is transported to the host nucleus where EBV finely regulates the expression of lytic genes. EBV lytic activation requires the coordinated expression of two viral immediate-early (IE) proteins: Zta, a leucine zipper DNA-binding protein (also called BZLF1, Zebra or EB1), and Rta protein (also called R) [49,50]. Zebra protein is a transcription factor that interacts with the Z response elements (ZREs) binding sites to activate the transcription of early genes required for viral DNA replication such as BALF5, BBLF4, BSLF1 and BMRF1 [51,52]. The replication complex amplifies the viral genome in a rolling-circle mechanism allowing the genesis of concatemers [53]. After replication, latent genes such as BGLF3, BDLF4, BVLF1, and BDLF3 are expressed to form structural proteins of the nucleocapsid, tegument and surface glycoproteins. These components form the viral particle which, when released to the outer host membrane, cause cell lysis [54].

## 4. Breast Cancer and EBV Epidemiology

EBV was the first human virus described as oncogenic and classified as carcinogen class I by the International Agency for Research on Cancer (IARC) [55]. Two epithelial tumors have been clearly associated with EBV infection: undifferentiated NPC which harbors EBV in almost 100% of cases and gastric cancer (GC) in almost 10% of cases. [56]. Additionally, EBV is strongly linked to malignancies such as Burkitt’s lymphoma (BL), Hodgkin’s lymphoma (HL), and T-cell lymphoma [57,58,59,60,61].

Interestingly, some lymphocyte-rich types of BC are morphologically similar to NPC [62,63]. There are well-documented publications that have evaluated the presence of EBV infection in BC [18,64,65]. However, the etiological role of EBV in BC is controversial, which may be attributable to different molecular techniques used, the type of clinical specimen, low expression of EBV proteins, and geographic or socioeconomic heterogeneity [66]. In 1995, Labrecque et al. first reported EBV presence in 19/91 (21%) BCs by polymerase chain reaction (PCR). Additionally, EBER-1 was detected in 6/19 (31.5%) BCs by in situ hybridization (ISH). Moreover, a non-statistically significant association with histopathological features was found [18]. Recent studies have also reported that between 30 and 50% of BCs are positive for EBV. Indeed, in 2001 Fina et al. collected 509 IDCs samples and found that 162/509 (31.8%) were EBV-1 positive, with no positive correlation among geographical areas or clinicopathological features [67,68]. In 2001, Chu et al. analyzed 48 BCs by multiple techniques and detected EBER-1 in 5/48 (10%) by ISH, EBNA-1 in 12/48 (25%) cases by IHC but failed to detect LMP-1 and ZEBRA proteins. Furthermore, EBNA-4 and LMP-1 gene fragments were detected by PCR in 7/48 (15%) cases. However, results were not conclusive because only a small subset of tumoral cells was EBER-1 positive and detection of prominent lymphocytic infiltration was only found in 5/7 (71%) PCR-positive cases [69]. In 2005, Kalkan et al. found EBV in 13/57 (23%) of BC tissues by PCR. However, EBV was found in 19/55 (35%) of control samples, suggesting no direct association between EBV and breast carcinogenesis [64]. Another study conducted in 2001 by Preciado et al. found EBV positivity in 12/39 (31%) cases by PCR. In addition, EBNA-1 was detected in 24/69 (35%) BCs by immunohistochemistry (IHC) [65]. Furthermore, in 2009 Joshi et al. detected EBNA-1 expression in 28/51 (54.9%) BC samples from Indian patients by IHC. The seroprevalence determined by EBNA-1 ELISA was higher in BCs (90.9%) than in the control group (81.8%) [70]. In 2010, Lorenzetti et al. evaluated the expression of EBV latent proteins by IHC in 71 BC biopsies classified as IDC and ILC. A higher positivity was observed in LMP-2A expression (16/22, 73%) than EBNA-1 (22/71, 31%), while LMP-1 was not detected. Additionally, EBER transcripts were also detected in 24/71 (31%) tissue samples by ISH [71]. In 2011, Mazouni et al. amplified the BAMH1C viral genomic region, finding 65/169 (33.2%) positive BCs. Furthermore, most EBV-positive tumors were estrogen receptor (ER)-negative, suggesting a more aggressive BC phenotype in those patients [72]. In 2011, Hachana et al. analyzed 123 Tunisian BC specimens and found EBV DNA in 33/123 (27%) cases. Neither EBER detection by ISH nor LMP-1 detection by IHC were positive. In 2011, Aguayo et al. performed the first study in Chile to analyze HPV and EBV infection in 46 BCs. EBNA-1 was found in 6.5% of BC tissues by RT-qPCR and HPV/EBV co-presence was detected only in 3/36 (2.1%) cases. A positive association between EBNA-1 and poor survival was statistically significant (*p* = 0.013) [73]. In 2012, Zekri et al. established EBV positivity in two Arab groups. The results were 18/40 (45%) and 14/50 (28%), respectively, though EBERs and LMP1 were not detected [74]. In 2013, a study conducted by Khabaz et al. analyzed 92 Jordanian BCs. PCR products for four EBV genes, EBER-2, EBNA-2, BNLF1 and gp220 showed 24/92 (26%) EBV-positive cases. Moreover, IHC for EBNA-1 showed 24/92 (26%) positivity as well. Interestingly, the EBV genome was also evident in 3/49 (36%) non-carcinomatous samples [19]. In 2014, Yahia et al. found LMP-1 and EBNA-4 expression in BCs in 49/92 (53.3%) and 10/92 (11%) cases, respectively. Interestingly, control tissues also harbored EBV genomes in 12/92 (24%) BCs. EBER positive signal was also detected in 18 BC biopsies by ISH. Moreover, methylation frequencies were evaluated reaching 84% in BRCA1 and BRCA2 BC susceptible genes [75]. In 2015, Richardson et al. found 25/70 (34.3%) of EBNA-1 positivity in 70 BCs from New Zealand and 9/70 (13%) in paired normal tissue, although EBV positivity was not associated with grade, receptor status, or disease stage [76]. In 2017, El-nabi et al. analyzed 42 BC specimens by nested PCR and found 12/42 (28.5%) positivity for EBNA-1 [77]. In 2017, Pai et al. found 25/83 (30.1%) EBER expression in primary invasive breast carcinoma (PIBC). A positive correlation between EBER-ISH positivity and clinicopathological features was found. Moreover, most BCs were classified as triple negative (56.5%) suggesting a worse prognosis in those patients [78]. In 2017, Fessahaye et al. analyzed 144 BC biopsies. The PCR for two genomic EBV regions showed presence of 40/144 (27.7%) even though control samples were EBV-negative. EBER signal was also detected in BC specimens by ISH in 5/14 (35.7%) cases and LMP-2A expression was 7/45 (15.5%) cases [79]. In 2019, Sharifpour et al. analyzed 37 BCs by nested PCR where 10/37 (27%) were positive for EBV DNA [80]. In 2020, Mofrad et al. found a 4/59 (6.7%) of EBNA-1 positivity in Iranian samples while all BC control samples were negative. Additionally, EBV-positive tumors were classified as high grade (II, and III) [81] (Table 1). 

Conversely, some studies have reported EBV absence in BCs. For instance, in 1998, Chu et al. analyzed 60 invasive BCs from Taiwanese patients and reported that all tissues were negative for EBV non-coding EBER-1, EBER-2 and EBNA-2 and LMP-1 oncoproteins when analyzed by IHC and ISH, respectively [82]. In 1988, Glaser et al. conducted a study on 107 BC samples collected from a diverse population group. The ISH analysis was unable to detect EBER-1 suggesting that EBV detection is geographically restricted to a subset of cases [83]. In 2001, Kijima et al. performed an extensive study in 761 patients clinically diagnosed with cancer. ISH failed to detect EBER in all 61 BC samples from Japanese patients. In 2002, Deshpande et al. analyzed 43 BC tissues and found neither EBERs presence by ISH nor nuclear positivity for EBNA-1, LMP-1 or LMP-2A by IHC, suggesting the absence of latency in BC cells [84]. In 2003, Herrmann et al. conducted a study in 59 BC samples, where only 4/59 (6.8%) were positive by PCR; however, all of these cases were EBERs-negative, suggesting the PCR findings derived from EBV infected lymphocytes detected in the tumor stroma. Additionally, EBNA-1 was not expressed in tumoral cells, concluding that EBV has an indirect role in BC pathogenesis [85]. In 2003, Murray et al. found that BC cells did not harbor EBER when analyzed by ISH. The very low number of EBV copies detected by RT-qPCR 19/92 (21%) did not derive from BCs [86]. Moreover, Perrigoue et al. detected a very low EBV DNA load in 45 BCs. Additionally, EBERs detection by ISH was negative suggesting that previous results reflected basal noise, leading us to conclude that EBV is not related to BC [87]. In 2011, Kadivar et al. failed to detect both EBV genomes by PCR and EBNA-2/LMP-1 by IHC, analyzing 100 BCs [88]. In 2019, Dowran et al. did not detect EBV in 150 BCs from Iran by BHRF1 gene fragment amplification [89] (Table 2).

## 5. EBV in Breast Cancer: Potential Mechanisms

The molecular mechanisms by which EBV may be involved in BC have not yet been established. However, the molecular characterization of breast tumors has shed some light on potential mechanisms. Thus, as occurs in other epithelial EBV-driven malignancies, EBNA1, LMP1, BZLF1, and BARF1 transcripts have been detected in BC specimens [90], suggesting a role for the corresponding proteins. Additionally, EBV-related tumors displayed a latency II type, characterized by EBNA1 and LMP1-2B expression accompanied by BXLF2 and BFRF3 lytic gene expression, suggesting that latent/lytic switch activation may be involved [91]. EBNA-1 contributes to episomal maintenance allowing the viral genome to tether to the host genome. Interestingly, EBNA-1 contributes to the oncogenic process through multiple mechanisms, including regulation of viral/cellular gene expression and evasion of the immune response (reviewed in [92]). LMP-1 is an oncogenic protein with functions such as the reprogramming of metabolism pathways, promotion of histone and DNA methylation, formation of extracellular vesicles, regulation of antiviral and antitumor immune responses, and promotion of chronic inflammation, among others (reviewed in [93]). BARF1 is an early lytic protein involved in immune evasion as well as contributing to carcinogenesis by promoting cell immortalization and anti-apoptotic effects and increasing cell proliferation (reviewed in [94]). 

Hu H et al. demonstrated that EBV infection of immortalized human mammary epithelial cells increased breast tumor formation when these cells were inoculated in NOD/SCID mice. Indeed, the authors concluded that EBV infection predisposes mammary epithelial cells to malignant transformation in cooperation with activated Ras but is no longer required once malignant transformation has occurred, consistent with a “hit and run” mechanism [91]. In line with this finding, in the TCGA database, a higher expression of the EBV epithelial receptor EphA2 was found in normal breast tissues than in BCs, suggesting that normal breast tissues are potentially more susceptible to EBV infection than breast tumors (Figure 2). 

It has also been reported that transfection of primate tissue explants with the p31 sub-fragment of EBV DNA results in immortalization of the human breast epithelial cells [96,97], although it remains unknown whether EBV infection occurs before the clonal expansion of breast epithelial cells or if previous DNA alterations are necessary to support latency, as suggested in NPC [98]. Jiun-Han L et al. have reported that HER2/HER3 activation is necessary for EBV-mediated breast cell transformation. Additionally, this effect is mediated by BARF0 expression, although the involved mechanism remains to be elucidated. In fact, BARF0 encodes a protein involved in immune evasion [99]. 

Yasui et al. (2001) suggested the hypothesis that delayed EBV infections is associated with an increased risk of developing BC [100]. Indeed, other EBV-associated diseases such as Hodgkin’s disease (HD) and multiple sclerosis have been associated with delayed EBV infection, clinically manifested as IM. It has been suggested that immune system overstimulation, which involves a cocktail of cytokine production and inflammation, may be involved in the effect of delayed EBV infection, promoting EBV-associated diseases such as BC.

Interestingly, exosomes released from EBV-infected B cells containing EBV miRNAs or LMP1 cargo can affect adjacent epithelial cells with a potential modulation of tumor behavior and immune evasion [101]. Thus, this possibility warrants more investigation in BC tissues which frequently exhibit tumor-infiltrating lymphocytes (TILs) [102]. Finally, we cannot deny the possibility of cooperation between EBV and other cofactors including xenobiotics and other viruses. For instance, Rajbongshi et al. (2021) reported that both EBV and LMP1 protein increase cell proliferation and induce α9-nAChR upregulation in BC cells. The authors proposed that EBV presence affects the response of BC cells to nicotine [103]. In addition, it has been shown that organophosphate pesticides such as malathion or parathion are related to BC by in vitro and in vivo approaches [104,105,106]. Notably, it was reported that chlorpyrifos, an organophosphate pesticide, can promote EBV lytic cycle activation by promoting BZLF1 expression by oxidative stress in a lymphoblastoid cell line. Considering the relevance of lytic activation in EBV-driven carcinogenesis [107], these issues lead us to speculate about the possibility of cooperation between EBV and pesticides for BC development. In addition, frequent viral coinfection has been detected in BCs including EBV, HPV and MMTV, suggesting the possibility of interactions among viral pathogens [108]. Finally, considering all these findings, a hypothetical model is suggested for EBV-mediated breast carcinogenesis, even though additional studies are necessary to solve this conundrum (Figure 3).

## 6. Conclusions and Remarks

The role of EBV infection in BC has not yet been established. However, epidemiological studies suggest an increased risk of BC in the presence of EBV. This association appears to be stronger in Asian countries when compared to European countries. Factors related to these differences have not been identified. Additionally, potential mechanisms involved in EBV-mediated breast carcinogenesis have been hypothesized. These mechanisms involve “hit and run” in which EBV infects primary breast cells though after infection the virus is no longer required for cancer progression. The molecular characterization of breast tumors identified viral products including EBNA-1, BZLF1 (Zta protein), BARF-1, BARF-0, BXLF-2, and BFRF-3, which suggest that an abortive lytic cycle may be involved. Additionally, we cannot deny the potential involvement of carcinogenic agents including HPV, MMTV, and xenobiotics (i.e., pesticides) and interactions among them. Additional studies are warranted to identify the mechanisms by which EBV may be related to BC, with a potential impact on prevention or treatment strategies. 

## Figures and Tables

**Figure 1 biology-11-00799-f001:**
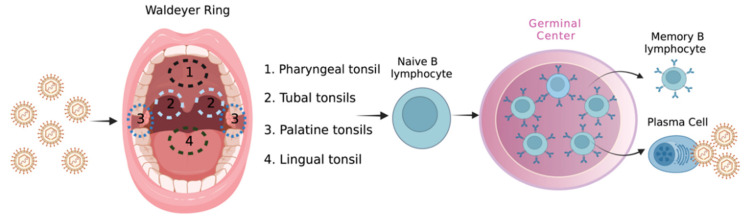
Model of primary infection by Epstein–Barr virus in the oropharynx.

**Figure 2 biology-11-00799-f002:**
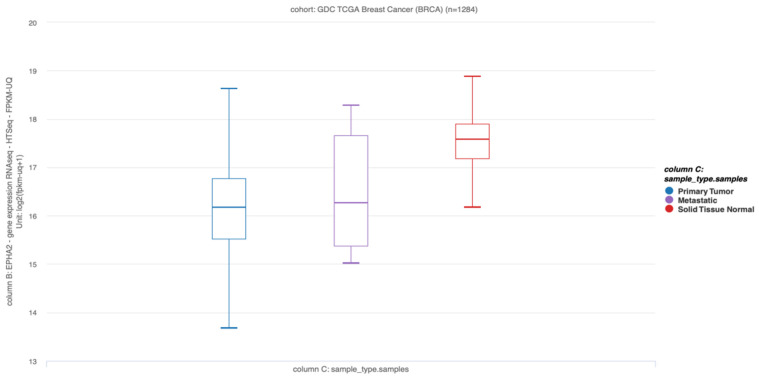
EphA2 transcripts in primary breast tumors, metastatic, and normal breast tissues (TCGA, n = 1284; *p =* 0.000, One-way ANOVA). Raw data were extracted from University of California, Santa Cruz (xena.ucsc.edu). UCSC Xena functional genomics explorer (https://xenabrowser.net accessed on 1 February 2022) [95].

**Figure 3 biology-11-00799-f003:**
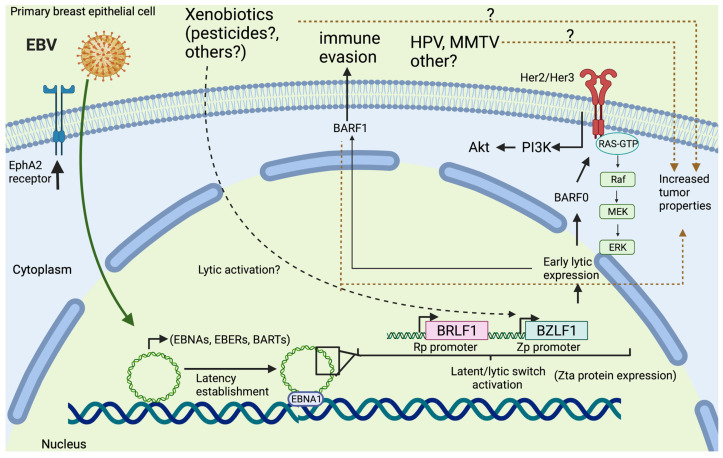
A hypothetical model of EBV role in breast carcinogenesis. Primary breast epithelial cells are susceptible to EBV infection, probably using EphA2 receptor. Once EBV latency is established, including EBNA-1, EBER and BART expression, the lytic switch can be activated. Xenobiotics may be involved in Zp activation as suggested in some models. BARF0 is involved in Her2/Her3 activation promoting tumor transformation by Ras/Raf/MEK/Erk and Pi3K/akt signaling pathways. BARF1 is involved in immune evasion and providing tumor properties. Xenobiotics or additional viral infections can cooperate with EBV for increasing breast tumor properties.

**Table 1 biology-11-00799-t001:** Epstein–Barr virus presence in breast cancer worldwide.

Author/Year	BC Type	EBV Detection	Detection Method
Labrecque 1995	IDC-DCIS BD/L-ILC MC-T/C	19/91 (21%)	PCR SB ISH
Fina 2001	IDC	162/509 (31.8%)	PCR SB ISH RT-qPCR
Chu 2001	IDC ILC	195/48 (10%)	ISH
Kalkan 2005	IDC-ILC NEC	13/57 (23%)	PCR
Preciado 2005	DC-LCI PC-AC	24/69 (35%)	PCR SB IHC
Joshi 2009	IDC-ILC NEC	28/51 (54.9%)	ELISA IHC
Lorenzetti 2010	IDC-ILC	22/71 (31%)	PCR IHC ISH
Mazouni 2011	IDC-ILC	65/196 (33.2%)	RT-qPCR
Hachana 2011	IDC-ILC MC	33/123 (27%)	PCR ISH IHC
Aguayo 2011	IDC-ILC CC	3/46 (6.5%)	RT-qPCR ISH IHC
Zekri 2012	IDC-ILC	18/40 (45%) 14/50 (28%)	PCR ISH IHC
Khabaz 2013	IDC-ILC MC-CC	24/92 (26%)	PCR IHC
Yahia 2014	IDC-ILC CIS	49/92 (53%)	PCR ISH
Richardson 2015	IDC	25/70 (34.3%)	qPCR
El-Nabi 2017	IDC	12/42 (28.5%)	Nested PCR IHC
Pai 2017	IDC-MPC ILC	25/83 (30.1%)	ISH
Fessahaye 2017	IDC-LC MC-NEC	40/144 (27.7%)	PCR ISH IHC
Sharifpour 2019	DC	10/37 (27%)	Nested PCR IHC
Mofrad 2020	IDC-ILC	4/59 (6.7%)	PCR

IDC, invasive ductal carcinoma; DCIS, ductal carcinoma in situ; BD/L, borderline ductal/lobular; ILC, invasive lobular carcinoma; MC, medullary carcinoma; CC, colloid carcinoma; T/C, tubular-cribriform; NEC, non-specified carcinomas; LCI, lobular carcinoma in situ; PC, papillary carcinoma; AC, adenocystic carcinoma; CIS, carcinoma in situ; MPC, metaplastic carcinoma; LC, lobular carcinoma; DC, ductal carcinoma; PCR, conventional polymerase chain reaction; SB, Southern blot; ISH, in situ hybridization; IHC, immunohistochemistry.

**Table 2 biology-11-00799-t002:** Epstein–Barr virus absence in breast cancer worldwide.

Author/Year	BC Type	EBV Detection	Detection Method
Chu 1998	IDC-ILC	0/60 (0%)	IHC ISH
Glaser 1998	BCs	0/107 (0%)	ISH
Kijima 2001	ADC	0/61 (0%)	ISH
Deshpande 2002	DC LC	0/43 (0%)	ISH IHC
Herrmann 2003	IDC-ILC MC NEC	0/59 (0%)	ISH IHC PCR
Murray 2003	DCIS-MC CC	0/98 (0%)	RT-qPCR ISH IHC
Perrigoue 2005	IDC-ILC TC-CC	0/45 (0%)	RT-qPCR ISH
Kadivar 2011	IDC-ILC AC-CRC PC-CC	0/100 (0%)	PCR IHC
Dowran 2019	DC-LC IDC-ILC	0/150 (0%)	PCR

IDC, invasive ductal carcinoma; ILC, invasive lobular carcinoma; BCs, breast carcinomas; ADC, a-denocarcinoma; DC, ductal carcinoma; LC, lobular carcinoma; MC, medullary carcinoma; NEC, non-specified carcinomas; DCIS, ductal carcinoma in situ; CC, colloid carcinoma; TC, tubular carcinoma; AC, apocrine carcinoma; CRC, cribriform carcinoma; PC, papillary carcinoma.

## Data Availability

Not applicable.

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
