# Peer review of "Epstein–Barr Virus Association with Breast Cancer: Evidence and Perspectives"

_biology, 2022, doi:10.3390/biology11060799_

Round 1

Reviewer 1 Report

The authors of the manuscript entitled “Epstein-Barr virus association with breast cancer: evidence and perspectives” reviews on possible association of EBV with Breast Cancer.  

Overall, the manuscript is well written with clear demonstration of figures and tables. The review shall represent a good addition to the increasing knowledge of EBV association with certain types of Breast Cancer.

I have no particular suggestion.   

Minor point:

Fig. 2: Lymphocyte is wrongly spelled.  

Page 7, Table 1: Information regarding EBV positivity in column 2 is wrong. The headline of Table 1 is missing.  

Author Response

Reviewer: The authors of the manuscript entitled “Epstein-Barr virus association with breast cancer: evidence and perspectives”reviews on possible association of EBV with Breast Cancer.  

Overall, the manuscript is well written with clear demonstration of figures and tables. The review shall represent a good addition to the increasing knowledge of EBV association with certain types of Breast Cancer.

I have no particular suggestion.   

Answer: Many thanks

Reviewer:  Minor point:

Fig. 2: Lymphocyte is wrongly spelled.  

Answer: The Figure 2 (In this new version is the Figure 1) was corrected

Reviewer: Page 7, Table 1: Information regarding EBV positivity in column 2 is wrong. The headline of Table 1 is missing.  

Answer: These mistakes were corrected. Many thanks.

Reviewer 2 Report

The relationship between EBV and breast cancer remains controversial. The authors tried to summarize epidemiological information and propose a possible model.

  1. In Table 2, the authors list the negative epidemiological findings in EBV-associated BC. Yet, many reports are ignored by the authors. For example, Chu et al 2001; Deshpande et al 2002; Hermann et al 2003, and Jacqueline G et al 2005, Dowran et al 2019.
  2. The authors mention genomic instability and clinical application in BC in page 3 and in Fig. 1 and Fig. 2. It is unassociated with the topic of this review. Also, EBV replication cycle in page 4 and Fig. 2 (page 5) are unrelated with the topic. The authors should focus on the relevance of EBV and BC.
  3. Line 343, wrong reference (101)
  4. Line 345-349, without reference
  5. The proposed model shows BZLF1 and BRLF1, which should be a bicistronic gene.

Author Response

Reviewer: The relationship between EBV and breast cancer remains controversial. The authors tried to summarize epidemiological information and propose a possible model.

In Table 2, the authors list the negative epidemiological findings in EBV-associated BC. Yet, many reports are ignored by the authors. For example, Chu et al 2001; Deshpande et al 2002; Hermann et al 2003, and Jacqueline G et al 2005, Dowran et al 2019.

Answer: We appreciate this observation. We have included the work of Deshpande et al 2002; Hermann et al 2003; Perrigoue J et al 2005 and Dowran et al 2019 as suggested. The additional Chu et al 2001 work suggested to be included in Table 2 (negative findings), was in fact added in Table 1 (positive findings) because EBER-1 was detected in 5/48 patients by ISH, which is the gold standard to determine EBV presence (EBV positive tumors highly express EBERs and this technique allows the discrimination of nuclear cell infection of lymphoid cell infiltrates). However, authors may have taken these results as no significant due that only a small subset of tumoral cells were EBER-1 positive (<1 in 1000 cells) in all 5 patients. Other methods used by the authors do not provide definitive evidence of EBV as they discussed.   Reviewer: The authors mention genomic instability and clinical application in BC in page 3 and in Fig. 1 and Fig. 2. It is unassociated with the topic of this review. Also, EBV replication cycle in page 4 and Fig. 2 (page 5) are unrelated with the topic. The authors should focus on the relevance of EBV and BC.

Answer: Many thanks for this comment. This section of genomic instability and clinical application was deleted from the manuscript. However, we consider that even though EBV replication cycle is not directly related with the topic “EBV in breast cancer”, the knowledge of this biological events is critical to understand the potential relationship between EBV and cancer. The final hypothetical model of breast cancer by EBV consider aspects of the viral replication cycle (entry, latency, etc).

Reviewer: Line 343, wrong reference (101)

Answer: This was corrected

Reviewer: Line 345-349, without reference

Answer: The reference was added.

Reviewer: The proposed model shows BZLF1 and BRLF1, which should be a bicistronic gene

Answer: This was corrected into the proposed model.

Reviewer 3 Report

Arias-Calvachi et al. in this review 'Epstein-Barr virus association with breast cancer: evidence and perspectives' summarize the data, in particular clinical one, about the association between EBV infection and BC. 

The review is well written and include most of the literature associated with this argument

The authors should add more informations about the EBV proteins involved in carcinogenesis and should also discuss more the importance of EBV lytic reactivation in this process. 

It is known that EBV is mostly a cofactor so authors could stress more this finding particularly in 'potential mechanism' and 'conclusion'. 

Lines 336. The authors have to mention the work on viral coinfection ( doi: 10.1080/21645515.2021.1975452)

Author Response

Reviewer: Arias-Calvachi et al. in this review 'Epstein-Barr virus association with breast cancer: evidence and perspectives' summarize the data, in particular clinical one, about the association between EBV infection and BC.

The review is well written and include most of the literature associated with this argument

Answer: Many thanks

Reviewer: The authors should add more information about the EBV proteins involved in carcinogenesis and should also discuss more the importance of EBV lytic reactivation in this process.

Answer: many thanks for this observation. Additional viral and cellular proteins were added into the manuscript and in the final model.

Reviewer: It is known that EBV is mostly a cofactor so authors could stress more this finding particularly in 'potential mechanism' and 'conclusion'.

Answer: many thanks for this observation, we added additional sentences to strengthen this issue in potential mechanism and conclusion sections.

Reviewer: Lines 336. The authors have to mention the work on viral coinfection ( doi: 10.1080/21645515.2021.1975452)

Answer: This work was cited and discussed.

Round 2

Reviewer 2 Report

I have no further comments.